# A Randomised Controlled Trial of Ice to Reduce the Pain of Immunisation—The ICE Trial

**DOI:** 10.3390/tropicalmed6030158

**Published:** 2021-08-28

**Authors:** Yashodha Ediriweera, Jennifer Banks, Leanne Hall, Clare Heal

**Affiliations:** Mackay Clinical School, College of Medicine and Dentistry, James Cook University, 475 Bridge Road, Mackay, QLD 4740, Australia; banks.jennifer.jcu@gmail.com (J.B.); leanne.hall@jcu.edu.au (L.H.); clare.heal@jcu.edu.au (C.H.)

**Keywords:** primary health care, cold therapy, analgesia, general practice, vaccinations

## Abstract

Background and objectives: vaccine injections are a common cause of iatrogenic pain and anxiety, contributing to non-compliance with scheduled vaccinations. With injection-related pain being recognised as a barrier to vaccination uptake in both adults and children, it is important to investigate strategies to effectively reduce immunisation pain. This prospective randomised controlled trial investigated the effects of applying an ice pack on vaccine-related pain in adults. Methods: medical students receiving the flu vaccination were randomised to receive an ice pack (intervention) or placebo cold pack (control) at the injection site for 30 s prior to needle insertion. Immediate post-vaccination pain (VAS) and adverse reactions in the proceeding 24 h were recorded. Results: pain scores between the intervention (*n* = 19) and control groups (*n* = 16) were not statistically significant (intervention: median pain VAS = 7.00, IQR = 18; control: median pain VAS = 11, IQR = 14 (*p* = 0.26). There were no significant differences in the number of adverse events between the two groups (site pain *p* = 0.18; localised swelling (*p* = 0.67); bruising *p* = 0.09; erythema *p* = 0.46). Discussion: ice did not reduce vaccination-related pain compared to cold packs. COVID-19 related restrictions impacted participant recruitment, rendering the study insufficiently powered to draw conclusions about the results.

## 1. Introduction

Vaccine injections are the most common cause of iatrogenic pain in childhood, and can cause anxiety in adults [1,2]. A 2003 survey reported 21% to 90% of adults experienced anxiety about pain associated with needle-based procedures [3]. In severe cases, this leads to injection phobia and subsequent syncopal attacks [4,5]. Vaccination pain is now a recognised adverse effect, and adequate pain management strategies should be incorporated into every vaccination [6].

The physiological mechanisms for analgesic effects of cold temperature are multitheoretical. The gate control theory of pain by Melzack and Wall in 1965 theorised those noxious inputs relayed by small, myelinated A-δ fibres and unmyelinated C fibres are inhibited by non-noxious stimuli concurrently conducted by A-β fibres to the dorsal horn. Consequently, the intensity of the ascending pain stimuli is decreased [7]. Kunesch et al. demonstrated that effects of skin cooling were more pronounced on C fibres compared to A fibres, leading to reduced autonomic response and pain sensation [8]. Another theory, based on animal models, is activation of the analgesic descending pathway stimulated by cold which produces opiate-like peptides [9]. In general, ice packs (<5 degrees Celsius) are known to reduce the temperature of skin and tissues to up 2 cm deep and induce almost immediate localised skin analgesia [10,11]. A previous study demonstrated that a 30 s application of ice reduced pain levels during tetanus vaccination in adults [12]. A 100C cold pack requires a minimum 20 min application to achieve a similar decrease in skin temperature, and therefore is unable to provide rapid skin analgesia [11].

Literature on use of cooling for injection-related pain is sparse, with studies showing conflicting results. The majority of cooling studies have investigated vapocoolants for their vaccination-related pain mitigation properties, with only three out of 13 randomised controlled trials (RCTs) using ice as a cooling method [13]. Only one of these was conducted in an adult population, [12] thus necessitating further research.

This prospective RCT investigates the effect of application of an ice pack (0 °C) to the site of injection prior to immunisation on flu vaccine-related pain levels. The secondary objective was to record adverse reactions to the cooling packs or flu vaccine.

## 2. Methods

### 2.1. Study Design and Participants

This multicentre, double-blinded prospective randomised controlled trial was registered with the Australian and New Zealand Clinical Trials Registry (ANZCTR)– ACTRN12621000064808. The published study protocol provides full methodological details. [14] This double blinded RCT was designed with involvement from the James Cook University Medical Student Association (JCUMSA). Medical students over the age of 18 years attending the annual JCUMSA flu vaccination clinics were invited to participate. Those with a known history of cold anaphylaxis, serious immunisation reactions or allergy to influenza vaccines were excluded. An independent research officer was responsible for recruitment and consenting of medical students to prevent potential coercion to participate. A detailed summary of the project, including potential risks, was provided and all participants gave written informed consent prior to the study. This research conforms to the Declaration of Helsinki and was approved by the James Cook University Human Research Ethics committee (H7871).

### 2.2. Randomisation and Blinding

Randomisation was performed at the participant level in a 1:1 ratio. The random sequence was generated from a computer-generated number table. Allocations were concealed using sealed, numbered, tamperproof opaque envelopes until participants were consented to the trial to minimise potential selection or confounding bias. Members of the research team involved in assessment or administration of vaccines had no role in the assignment process. Participants were blinded to treatment allocation. Ice packs and placebo cold packs were stored separately in identical coolers to blind the doctor administering vaccines.

### 2.3. Intervention Procedure

All participants had an ice pack (intervention), or cold pack (control) applied to the site of injection for 30 s prior to the administration of the influenza vaccine. A vaccination protocol, modelled on the Royal Australian College of General Practitioners (RACGP) immunisation and influenza prevention guidelines, was used to standardise management across both study arms [15,16]. Influenza vaccines were administered immediately after removal of the ice or cold pack and all participants received a standard set of verbal and written post-immunisation instructions.

Modelled on previous studies investigating adult vaccination pain, our study utilised a 100 mm Visual Analogue Scale (VAS) for measuring post-immunisation pain [12,17]. A 15 mm difference on the VAS was selected as the minimum clinically significant difference for our study [18]. Further, the VAS has a standard deviation of 26 mm. As such, a sample size of 45 per group was calculated to achieve 80% power with *p* = 0.05 [19].

### 2.4. Data Collection

Baseline data included age, sex, ethnicity and past medical history. Participants recorded their immediate post-immunisation pain on a 100 mm VAS with 0 mm being no pain and 100 mm being the highest pain [18,19,20].

Secondary outcome measures included adverse reactions to the vaccination process, such as anaphylaxis, skin irritation, redness or contact dermatitis. Participants were advised to assess the injection site 24 h post-vaccination, complete a self-assessment form and return it to the clinic within 48 h of vaccination. Additionally, the vaccination sites were to be assessed for evidence of side effects if patients present opportunistically or for review due to side effects.

### 2.5. Statistical Analysis

The primary analysis was by intention-to-treat of all randomised participants. The individual person was considered as the unit of analysis. Significance was considered *p* < 0.05. Baseline data across the two groups were assessed for marked differences. Numerical data were described as median and interquartile range (IQR). Pain VAS was compared using independent samples median test and Mann–Whitney U test. Secondary outcomes were analysed using two-tailed Fisher’s exact probability testing. Data were analysed using IBM SPSS Statistics software [21].

## 3. Results

Forty participants were assessed for eligibility, five of which could not attend the vaccination clinic due to flu-like symptoms or recent travel from high risk COVID-19 areas. Ideally the flu vaccination should be timed such that peak protection is afforded at the peak of the flu season. Due to COVID-19 restrictions and the uncertainty of lockdowns, the flu clinic was delayed and peak protection would have fallen outside the usual flu season, therefore the study was unable to be continued until the sample size was met. Thirty-five patients were randomised with an allocation ratio of 1:1 and included in the final analysis: 19 in the intervention group (ice packs) and 16 in the control group (cold packs) (Figure 1). No patients were lost to follow up, and intention-to-treat and per protocol analysis were equivalent. There were no protocol violations.

There were no differences at baseline between the intervention and control groups (Table 1).

### 3.1. Primary Outcome

Data were not normally distributed (*p* = 0.001, two-tailed Shapiro–Wilk test). The distribution of pain scale scores in both groups were positively skewed (intervention group = 0.55; control group = 1.32). As demonstrated in Table 2, the median pain VAS scores did not differ between the groups (intervention group: median pain VAS = 7.0, IQR = 18; control group: median pain VAS = 11.0, IQR = 14; *p* = 0.26, two-tailed Mann–Whitney U test).

### 3.2. Secondary Outcomes

Localised pain 24 h post-immunisation was experienced by 42.1% in the intervention group and 68.8% in the control group (*p* = 0.18, two-tailed Fisher’s exact probability test). Localised swelling was reported by 21.1% in the intervention group and 12.5% in the control group (*p* = 0.67, two-tailed Fisher’s exact probability test). Of those in the control group, 18.8% reported bruising and 6.3% reported erythema while neither of these adverse events were reported in the intervention group (Table 3).

## 4. Discussion

Patients who received ice packs reported lower median pain scores compared to those given cold packs, however the results were not significant, potentially due to under powering of the study. There was no difference in adverse effects such as redness, swelling, bruising of the skin or prolonged pain, localised to the site of injection.

The subjective nature of pain makes it difficult to measure [22]. Individual characteristics including age, sex, past medical history and previous pain experiences are known to affect the perception of pain [22]. Medical students were recruited as participants because of convenience in organising the study around student-driven influenza clinics. As health literate participants, students were more likely to be compliant, and more reliable in the reporting of pain compared with the normal general practice population, however they may also be less likely to be anxious, and their perception of pain may differ as reflected by low VAS scores.

Ice is available in both primary care and hospital settings and has a quick onset of action, approximately 30s [12]. In addition, compared to topical local anaesthetic creams (approximately AUD 10 per dose) and vapocoolant sprays (AUD 0.70 per dose), ice packs (AUD 0.50 per dose) are more cost effective [1,23]. Ice packs have few associated adverse effects. Cold-induced anaphylaxis is the most severe potential side effect of ice, but it is rare and there are currently no conclusive data regarding its incidence [24]. The main disadvantage is their short life span in ambient temperature.

Injectable vaccines, administered according to standard guidelines, often vary in degree of painfulness depending on their composition [25]. In children, pneumococcal conjugate vaccine (PCV) is considered more painful compared to the diphtheria, tetanus and acellular pertussis (DTaP) vaccine and *Haemophilus influenza* type b (Hib) vaccine [26]. Likewise, concurrent hepatitis A and B vaccinations in adults demonstrated increased levels of pain compared to hepatitis B vaccination alone [27]. The current trend towards increased number of recommended vaccines, further makes the use of readily available and inexpensive ice packs as an analgesic agent attractive to enhance acceptance of vaccine injections.

## 5. Limitations

Compliance with government regulations, implemented during the COVID-19 pandemic, was required to ensure the safety of patients, health professionals and research staff. Some members of our research team and participating GP practices were deployed to other areas, with different responsibilities. Strict travel restrictions prevented travel to one of our clinical research centres in Townsville. As such, recruitment was from a single centre in Mackay, and our sample size was considerably reduced. Government restrictions permitted a maximum of 10 people in the participating general practice at one time and several rooms were required to ensure adequate social distancing. To comply with quarantine guidelines, students were advised not to attend the clinic if they were symptomatic or had recently travelled to high-risk COVID-19 locations. Our final sample size of 35 participants fell short of the 90 required to reach the desired study power, thereby limiting the potential to detect if ice packs were able to reduce pain associated with vaccine injection. Another limitation of this study was that participants were not asked to report their fear of needles from past experiences as these could potentially impact the pain ratings for the current injection. Given the relatively low VAS scores for vaccination-related pain in this study, it is unlikely that the variable of anxiety from previous vaccinations impacted the results.

## 6. Conclusions

We hypothesised that pain scores would be lower with the use of ice; however, results were inconclusive, potentially due to under-powering. Future studies investigating the efficacy of ice in reducing vaccine-related pain with larger sample sizes could potentially inform current practice.

## Figures and Tables

**Figure 1 tropicalmed-06-00158-f001:**
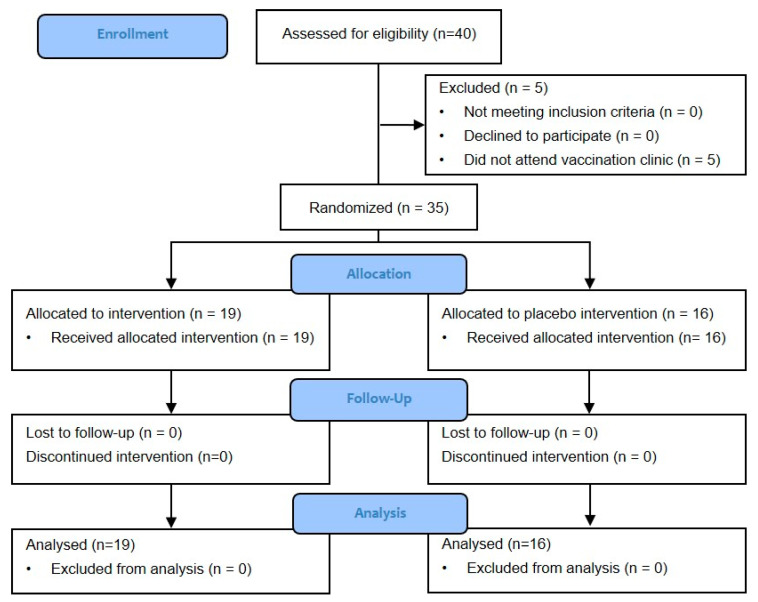
CONSORT flow diagram.

**Table 1 tropicalmed-06-00158-t001:** Baseline characteristics for participants included in the final analysis.

Characteristic	Intervention Group *n* = 19(%)	Control Group *n* = 16 (%)
Median age in years (IQR)	23.00 (3)	22.50 (1)
Gender		
Female	8 (42.1%)	12 (75%)
Male	11 (57.9%)	4 (25%)
Ethnicity		
Aboriginal and Torres Strait Islander	0	0
Caucasian	9 (47.4%)	7 (43.8%)
Other	10 (52.6%)	9 (56.3%)

**Table 2 tropicalmed-06-00158-t002:** Immediate post-immunisation pain scored on VAS.

Pain VAS (cm)	Intervention Group (*n* = 19)	Control Group (*n* = 16)	*p*-Value
Median (min–max)	7 (0–26)	11 (0–48)	0.26
IQR	18	14	

**Table 3 tropicalmed-06-00158-t003:** Adverse effects 24 h post-immunisation.

Adverse Event	Intervention Group N = 19 *n* (%)	Control Group N = 16 *n* (%)	*p*-Value
Localised to site of injection			
Pain	8 (42.1%)	11 (68.8%)	0.18
Swelling	4 (21.1%)	2 (12.5%)	0.67
Bruising	0 (0%)	3 (18.8%)	0.09
Redness	0 (0%)	1 (6.3%)	0.46
Itching	0 (0%)	0 (0%)	N/A
Systemic			
Anaphylaxis	0 (0%)	0 (0%)	N/A

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
