# Peer review of "A Randomised Controlled Trial of Ice to Reduce the Pain of Immunisation—The ICE Trial"

_tropicalmed, 2021, doi:10.3390/tropicalmed6030158_

Round 1
Reviewer 1 Report
Thank you for the opportunity to peer review the paper titled “A randomised controlled trial of ice to reduce the pain of immunisation – the ICE trial” submitted to Tropical Medicine and Infectious Disease.
The introduction clearly outlines the importance of the study.
Although the study protocol is In Press, please state if the protocol was also registered, and if so, include the registry and number.
Methods:
Include an explanation about sample size. It is stated that plans were to enrol 45 patients in each group, but state what this was based on. Was a power calculation performed? What was the anticipated difference in primary outcome of VAS scores?
State the explanations given for the VAS score for the participants.
Results:
Unfortunately for the investigators, recruitment could not occur as planned, and the sample size of only 35 is well below the 90 planned. However this has been well described and limitations of results considered. Suggest adding an explanation as to why the study could not continue for longer. Was this based on funding, or was this a student project? This needs to be made clear.
Change ‘subjects’ to participants
For Table 1, as there were no Aboriginal or Torres Strait Islanders enrolled, suggest to remove this category. As > 50% were ‘other’, please report further details.
Discussion:
Re-iterate that the VAS scores were low. Although it is stated that medical students are less likely to be anxious, and their perception of pain may differ, it is important here to note the very low scores. Include a statement abut the flu vaccination in general.
Include a limitation about not ascertaining previous fears of needles. Although no sub-analyses could have been done on this factor, given the sample size it is important to consider.
Reference list:
Review reference list for accuracy. Some journal titles italicised, and some are not.
The link to the RACGP immunisation guidelines is not live.
Author Response
We thank the reviewer for their time in reviewing our paper. We have made revision reflecting the points raised by the reviewer. Please see the attached document.

Reviewer 2 Report
This is a well-designed and well-reported study based on a hypothesis that pain scores would be lower with the use of ice at the vaccination site. Findings showed no significant difference with and without use of ice on pain scores. Considering the limitations of the study, which are described by the authors, the negative findings from this study must be cautiously interpreted, and a larger sample size study can test will no effect be the true finding or will be changed. Practical considerations for this study might not be similar for other setting or in the future considering the current pandemic conditions. However, the hypothesis will stay valid for further investigation. Age, sex, previous experience, and expectation of participants might play a role. In addition, finding the mechanistic aspects of the effect or lack of effect can be evaluated. Other measures besides pain can be added for example sensitivity to thermal or mechanical stimuli in response to von frey hair or thermal probes. This can be done by aid of QST bedside tests to provide further clarification on A delta or C fibers and nociception properties before and following the administration of ice. The application time and form can also be evaluated to identify an optimal application.
Author Response
We thank the reviewer for these comments and methodological suggestions for future studies. Our recent review concurs with your views regarding optimal application time and type.